# Physicochemical Stability of a Novel Tacrolimus Ophthalmic Formulation for the Treatment of Ophthalmic Inflammatory Diseases

**DOI:** 10.3390/pharmaceutics14010118

**Published:** 2022-01-04

**Authors:** Marion Barrieu, Philip Chennell, Mouloud Yessaad, Yassine Bouattour, Mathieu Wasiak, Mireille Jouannet, Yoann Le Basle, Valérie Sautou

**Affiliations:** 1CHU Clermont-Ferrand, Pôle Pharmacie, F-63003 Clermont-Ferrand, France; mbarrieu@chu-clermontferrand.fr (M.B.); myessaad@chu-clermontferrand.fr (M.Y.); mwasiak@chu-clermontferrand.fr (M.W.); mjouannet@chu-clermontferrand.fr (M.J.); 2Université Clermont Auvergne, CHU Clermont Ferrand, Clermont Auvergne INP, CNRS, ICCF, F-63000 Clermont-Ferrand, France; ybouattour@chu-clermontferrand.fr (Y.B.); ylebasle@chu-clermontferrand.fr (Y.L.B.); vsautou@chu-clermontferrand.fr (V.S.)

**Keywords:** tacrolimus, ophthalmic solution, physicochemical stability, container-content interaction, leachable compound

## Abstract

Tacrolimus is an immunosuppressant used to treat a large variety of inflammatory or immunity-mediated ophthalmic diseases. However, there are currently no commercial industrial forms available that can provide relief to patients. Various ophthalmic formulations have been reported in the literature, but their stability has only been tested over short periods. The objective of this study was to evaluate the physicochemical stability of a preservative-free tacrolimus formulation (0.2 and 1 mg/mL) at three storage temperatures (5 °C, 25 °C and 35 °C) for up to nine months in a multidose eyedropper. Analyses performed were the following: visual inspection and chromaticity, turbidity, viscosity, size of micelles, osmolality and pH measurements, tacrolimus quantification by a stability-indicating liquid chromatography method, breakdown product research, and sterility assay. In an in-use study, tacrolimus quantification was also performed on the drops emitted from the eyedroppers. All tested parameters remained stable during the nine month period when the eyedrops were stored at 5 °C. However, during storage at 25 °C and 35 °C, several signs of chemical instability were detected. Furthermore, a leachable compound originating from a silicone part of the eyedropper was detected during the in-use assay. Overall, the 0.2 mg/mL and 1 mg/mL tacrolimus ophthalmic solutions were physicochemically stable for up to nine months when stored at 5 °C.

## 1. Introduction

Tacrolimus (TAC), also named FK-506, is a macrolide immunosuppressant of the calcineurin inhibitor pharmacological class that binds to a specific cytosolic protein FKBP12 immunophilin (also named FK506 binding protein) [1,2], forming a complex that binds to calcineurin and inhibits it. Inhibiting calcineurin blocks the dephosphorylation of NFAT (nuclear factor of activated T cells), preventing it from crossing the nuclear envelope and entering the nucleus. Therefore, NFAT will not bind to gene promoter regions, thus decreasing the transcription of cytokines like IL-2 which have an essential role in T cell activation [3]. In addition to use in the field of transplantation to prevent graft rejection, TAC is also used to treat a large variety of inflammatory or immunity-mediated ophthalmic diseases such as dry eye syndrome (caused by Sjögren’s syndrom [4] or by graft versus host disease (GVHD) [5]), corneal or conjunctive immunological diseases [6], chronic follicular conjunctivitis or severe vernal and atopic keratoconjunctivitis [7,8,9]. Because of its mechanism of action, which is similar to cyclosporine, and its strong immunosuppressive power, tacrolimus is a good candidate to manage ophthalmic pathologies [6] for which cyclosporine is not tolerated or no longer effective. Moreover, it expands the therapeutic arsenal, particularly for vernal keratoconjunctivitis (VKC) and allergic eye diseases, the management of which is complex. According to the literature, ophthalmic TAC is used at various concentrations, depending on the disease, varying from 0.003% to 0.1% (*m/v*). Indeed, tested concentrations generally ranged from 0.003% to 0.03% for the treatment of VKC, and from 0.02% to 0.1% for the treatment of allergic eye diseases [10,11]. The administration can fluctuate from once to several times a day according the concentration and for a duration of treatment which can range from a few days to several months or years for chronic diseases [8].

Tacrolimus is a hydrophobic molecule with a calculated log P of 2.7 [12]. Thus, its low solubility in aqueous solutions at clinically relevant concentrations makes the development of a stable ophthalmic formulation complex. To date, there is no tacrolimus-based ophthalmic medication commercially available that is well tolerated and that can provide relief to patients. Talymus, which is a 1 mg/mL (0.1%) tacrolimus suspension, has been marketed since 2008 in Japan, but is not available in USA or in Europe, except in France under a special nominative temporary use authorization [13]. However, this medication can also cause numerous adverse effects such as burning or foreign body sensation in the eye (40% of patients), eye irritation (more than 20% of patients) eye pain, ocular hyperaemia, photophobia, etc. [14]. In addition, and despite the fact that medications from the treatment of chronic ophthalmic diseases are ideal candidates for preservative-free formulations [15], Talymus contains a preservative (benzalkonium chloride which is a quaternary ammonium compound), which is well known to cause cytotoxic damage to conjunctival and corneal epithelial cells, resulting in signs and symptoms of ocular surface disease, at concentrations starting at 0.005% [16]. Whilst the concentration of benzalkonium chloride in Talymus is not disclosed in the available Product Information, commonly used concentrations in ophthalmic medications range between 0.004 and 0.025%, and numerous studies have shown that eye drops with this preservative cause sometimes severe clinical consequences for the patients, comparatively to unpreserved medications [16,17]. This has led to numerous formulations being developed and tested by the pharmaceutical and medical communities. Some ophthalmic formulations based on castor oil [18] or olive oil [19] have been reported in the literature, but numerous adverse effects such as redness, burning, itching have been identified due to the use of these oily excipients. Other formulations have been tested, but their long-term stability (more than three months) is either insufficient [20] or was not evaluated [21]. These galenic and chemical issues make it complicated for compounding pharmacies to develop and produce tacrolimus ophthalmic solutions. The objective of this study was therefore to develop a physicochemically stable, preservative-free formulation of tacrolimus, at two different concentrations, 0.2 mg/mL (0.02% *m/v*) and 1 mg/mL (0.1% *m/v*), in order to cover a wide therapeutic spectrum of ophthalmic illnesses (VKC and allergic eye diseases).

## 2. Materials and Methods

### 2.1. Preparation and Storage of TAC Formulation

Tacrolimus ophthalmic solutions were prepared at two concentrations: 0.2 mg/mL (0.02% *m/v*) and 1 mg/mL (0.1% *m/v*). Tacrolimus powder was dissolved into absolute ethanol at room temperature under gentle agitation. The Kolliphor EL^®^ (KEL) was then added and mixed under vigorous mechanical agitation. In order to stabilize the pH, the hydrogenophosphate–hyaluronate buffer solution was added. The quantities used are summarized in Table 1. The final concentrations of ethanol were of 3.76 mg/mL (0.08 µmol/mL) and 18.79 mg/mL (0.41 µmol/mL) for, respectively, the 0.2 and 1 mg/mL tacrolimus formulation.

The final solutions of tacrolimus were filtered through a 0.22 µm filter Stericup^®^ Sterile Vacuum Filtration Systems (Merck Millipore, MC2, Clermont-Ferrand, France) and then an aseptic repartition was realized under the laminar airflow of an ISO 4.8 microbiological safety cabinet using a conditioning pump (Repeater^®^ pump, Baxter, Guyancourt, France). The low-density polyethylene multidose eyedroppers (reference VPL28B10N02, Laboratoire CAT, Lorris, France), possessing a sterility preserving system (Novelia^®^ nozzle) using silicones parts to hermetically release the eye drops, were filled with 5 mL and stored until analysed.

### 2.2. Study Design

The stability of the tacrolimus formulations was studied in unopened eyedroppers stored in the dark vertically for nine months at different storage temperatures: 5 ± 3 °C (Liebherr refrigerator), at 25 ± 1 °C and 35 ± 1 °C with 60 ± 5% residual humidity in a validated climate chamber (Binder GmbH, Tuttlingen, Germany). A simulated used study was also performed during 28 days at 5 °C after 8 months of storage at 5 °C for both concentrations.

### 2.3. Stability of Tacrolimus in Unopened Multidose Eyedroppers

Immediately after the preparation (Day 0 = D0) and at months 1 (M1), 2 (M2), 3 (M3), 4 (M4), 6 (M6), 8 (M8) and 9 (M9), four units (*n* = 4) of each concentration and each storage temperature condition were taken for analysis. The different parameters that were evaluated during the study are summarized in Table 2. At the initial day (D0), the analyses were performed immediately after conditioning in order to have the most representative results of the initial conditions (with the least degradation or modification of parameters).

### 2.4. Stability of Tacrolimus in Opened Multidose Eyedroppers (Simulated Use Study)

A simulation of patient use was performed after eight months of storage at 5 °C. Eight units of each concentration were opened and one drop from each eyedropper was manually emitted out of the bottle and collected for analysis every day (except weekends) twice a day (morning and evening) for one month and tacrolimus concentration was measured in the emitted drops. To ensure that a sufficient volume was collected to allow tacrolimus quantification, at each analysis time the drops were pooled by two (each time from the same two vials, which remained associated throughout the analysis). Thus, the results of these analyses were returned with *n* = 4. The units were stored vertically at 5 ± 2 °C between drops gathering. Moreover, after the 28 days, units were opened and remaining solution was subjected to tacrolimus quantification as well as pH and osmolality measurements.

### 2.5. Analyses Performed on the Tacrolimus Solution

#### 2.5.1. Visual Inspection

The multidose eyedroppers were emptied into glass test tubes without passing through the delivery device, and the tacrolimus solutions were visually inspected under daylight and under polarized white light from an inspection station (LV28, Allen and Co., Liverpool, UK). Aspect and colour of the solutions were noted, and a screening for visible particles, haziness, or gas development was performed.

#### 2.5.2. Chromaticity and Luminance Analysis

Chromaticity and luminance were measured with a UV-visible spectrophotometer (V670, Jasco Corporation^®^, Lisses, France) using the mode Colour Diagnosis of the built-in software (Spectra Manager^®^, version 2.12.00, Jasco Corporation^®^, Lisses, France) with a quartz measuring cell. The CIE L*a*b* colour parameters were used to represent the colour changes [22,23]. Transmittance spectra were obtained by using a light source D65 (wavelength between 780 and 380 nm), data pitch 5 nm, colour matching JIS Z8701-1999 and scan speed 400 nm/min. The difference of colour perception (∆E) was calculated using the following equations for the spectrophotometric analysis [24]:∆a= a* − a_0_*(1)
∆b*= b* − b_0_*(2)
∆L*= L* − L_0_*(3)
∆E = ((∆L*)² + (∆a*)² + (∆b*)²)/0.5(4)
a0*, b0* and L0* were the initial values at day 0 and ∆a*, ∆b* and ∆L* were the difference in chromatic coordinates and lightness. The values of each colour parameter are expressed as the mean of four different samples.

#### 2.5.3. Tacrolimus Quantification and Breakdown Products (BP) Research 

Chemicals and instrumentation

The quantification method used was directly transposed from the stability indicating ultra-high performance liquid chromatography (UHPLC) method published by Peterka et al. [25], after minor adaptations from UHPLC to high performance liquid chromatography (HPLC). The HPLC separation column used was a Kinetex^®^ Core-Shell 2.6 µm EVO C18 100 Å, 100 mm × 3 mm, reference 00D-4725-Y0, (Phenomenex, Le Pecq, France), with an associated security guard ULTRA EVO-C18 sub2 µm-Coreshell reference AJ0-9296 (Phenomenex, Le Pecq, France). This column has an equivalent stationary phase to that used by Peterka et al. [25] (C18 phase, USP L1 classification). Due to the modification of the column granulometry, the flow rate was increased to 1 mL/min and the mobile phase gradient was lengthened. The method was validated on a reverse-phase HPLC Prominence-I LC2030C 3D with diode array detection (Shimadzu France SAS, Marne La Vallée, France), and the associated software used to record and interpret chromatograms was LabSolutions^®^ version 5.82 (Shimadzu France SAS, Marne La Vallée, France). The mobile phase was a gradient mixture of phases A and B. Mobile phase A was an aqueous ortho-phosphoric acid solution at 85% (0.1 %, *v/v*). Mobile phase B was prepared by mixing 500 mL of ACN (Acetonitrile for HPLC, 99.95% ref 412342 2.5L) and 47 mL of MTBE (tert-Butyl methyl ether for HPLC, ≥99.8%, Sigma Aldrich, Saint Quentin Fallavier, France), ref 34875-1L). The column oven was set at 70 °C and the mobile phase was stored at ambient temperature before reaching the column. The gradient used is presented in Table 3. In order to verify the correct transposition of the method and to allow the identification as proposed in the original method of tacrolimus and its equilibrium compounds I and II, related substances and byproducts (ascomycin) and breakdown products (tacrolimus alpha-hydroxy acid and tacrolimus regioisomer), the relative retention times of tacrolimus and ascomycin (impurity A) were checked against those obtained by Peterka et al. [25]. These compounds were analysed as recommended by the tacrolimus European Pharmacopoeia monography [26].

Method validation

Linearity was verified by preparing one calibration curve daily for three days using five concentrations of tacrolimus (European Pharmacopoeia reference standard Y0001925) at 2.5, 5, 20, 50 and 100 µg/mL, diluted with a solvent consisting of ACN-water (70:30, *v/v*). Each calibration curve was considered acceptable if it had a determination coefficient R^2^ equal or higher than 0.999. Variance homogeneity of the curves was verified using a Cochran test. ANOVA tests were applied to determine the applicability of the linear model. Each day for three days, six solutions of tacrolimus 30 µg/mL were prepared, analysed, and quantified using a calibration curve prepared the same day [27]. To verify the method precision, repeatability was estimated by calculating relative standard deviation (RSD) of intraday analysis and intermediate precision was evaluated using an RSD of interdays analysis [27]. Both RSDs were considered acceptable if they were lower than 5%. Method accuracy was verified by evaluating the recovery of five theoretical concentrations to experimental values found using mean curve equation, and results were considered acceptable if found within the range of 90–110%. The overall accuracy profile was constructed according to Hubert et al. [28,29,30]. An evaluation of the matrix effect was performed by reproducing the previous methodology in the presence of all excipients included in the formulation and by comparing the calibration curves and intercepts. Specificity was also assessed by comparing the UV spectra obtained with the DAD detector for tacrolimus with and without the excipients.

#### 2.5.4. pH 

For each unit, pH measurements were made using a Seven Multi TM pH-meter with an In Lab Micro Pro ISM glass electrode (Mettler-Toledo, Viroflay, France). Measures were preceded by an instrument validation using standard buffer solution of pH 1.68, pH 4.01, pH 7.01 and pH 10.01 (Mettler-Toledo, Viroflay, France).

#### 2.5.5. Osmolality

The osmolality was measured on 20 µL samples using a freezing point osmometer Model 2020 Osmometer^®^ (Advanced instruments Inc., Radiometer, SAS, Neuilly Plaisance, France). Measurements were preceded by an instrument validation performed using a calibrated osmolality standard of 290 mOsmol/kg.

#### 2.5.6. Turbidity 

Turbidity of the different solutions was measured using a 2100Q Portable Turbidimeter (Hach Lange, Marne La Vallée, France). In order to obtain the necessary volume for each analysis (>15 mL), four samples per analysed experimental condition and assay time have to be pooled. The results were expressed in Formazin Nephelometric Units (FNU).

#### 2.5.7. Viscosity Measurements

Viscosity was measured using a Brookfield DV1 viscosimeter Labomat associated with a circulation thermostat Julabo, at 25 °C. In order to obtain the necessary volume for each analysis (>20 mL), four samples per analysed experimental condition and assay time have to be pooled. The results were expressed in cP.

#### 2.5.8. Micelle Size Measurements

The size of the tacrolimus micelles was determined by dynamic light scattering measurements using a Zetasizer Nano ZS (Malvern Instruments SARL, Orsay Cedex, France). Each sample (1 mL conditioned a in clear disposable polystyrene cell) was automatically screened 3 times and the results were then averaged. The size distribution by intensity was converted into a size distribution by volume by the instrument’s software after use of the solution’s refractive index as indicated by the user manual. Comparatively, samples of each formulation containing just the excipients (no tacrolimus) were also submitted to this analysis.

#### 2.5.9. Sterility Assay

The sterility of the solutions was assessed as indicated in the European Pharmacopoeia sterility assay (2.6.1) [31]. Under the laminar air flow of an ISO 4.8 microbiological safety cabinet, each analysed eyedropper was opened, and its contents were filtered under vacuum using a Nalgene^®^ analytical test filter funnel onto a 47 mm diameter cellulose nitrate membrane with a pore size of 0.45 mm (ref 147-0045, Thermo Scientific, purchased from MC2, Clermont-Ferrand, France). The membranes were then rinsed with 500 mL deionized water (Versylene^®^; Fresenius Kabi, Louviers, France) and divided into two equal parts. Each individual part was transferred to each of a growth medium of fluid thioglycolate and soya trypcase. Each culture medium was then incubated for 14 days at 30–35 °C (thioglycolate broth) or 20–25 °C (soya trypcase broth) and visually examined for any signs of microbial growth.

#### 2.5.10. Determination of the Volume of an Eye Drop

To determine an average density, 200 µL of the prepared solution was weighed ten times. Then ten drops emitted from the eyedroppers were weighed six times to determine the volume of one drop from the previously determined density. This operation was performed for both concentrations of eye drops.

### 2.6. Degradation Kinetics during Storage

In order to be able to estimate the potential impact of temperature excursions during storage, an evaluation of the degradation kinetics was investigated. Because the study was performed with three different temperatures (5 °C, 25 °C, and 35 °C), it was possible to evaluate the value of the reaction coefficient for an unstudied temperature using the Arrhenius equation [32]:(5)$$k=A\times \exp\left(-\frac{\mathrm{Ea}}{RT}\right)$$

With k the reaction rate coefficient, A the pre-exponential factor, Ea the activation energy, R the universal gas constant = 8.314 J·mol^−1^·K^−1^ and *T* the temperature in Kelvin.

First, the degradation rate k for each storage temperature was calculated after verification that the concentration (C) decreased following a first order reaction (i.e., C = C_0_ e^−kt^), by plotting Ln(C) as function of the time. Then, each Ln(k) value that was obtained was plotted as a function of 1/*T*. The curve has a slope value equal to −Ea/R and an intercept of Ln(A):(6)$$\mathrm{Lnk}=\mathrm{LnA}-(\frac{\mathrm{Ea}}{R}\times \frac{1}{T})$$

It is then possible to extrapolate, for different temperatures, the k value to determine a stability time such as the time (*t*) for which the concentration declined below 90% of the initial concentration, for this given different temperature. Once the new k is obtained for a given temperature, it is injected into the integrated equation of a reaction of order 1 and *t* determined using the following equation: (7)$$\mathrm{Ln}\left(C0\times 0.9\right)=\mathrm{LnC}0-(k\times t90)\;$$

Which means *t* = −Ln 0.9/k.

### 2.7. Data Analysis—Acceptability Criteria

The stability of the tested tacrolimus formulations was evaluated using the following physicochemical parameters: visual aspect of the solution, turbidity and micelle size, viscosity, pH and osmolality, tacrolimus concentration and a research of BPs. The study was conducted following methodological guidelines issued by the International Conference on Harmonisation for stability studies [33] and recommendations issued by the French Society of Clinical Pharmacy (SFPC) and by the Evaluation and Research Group on Protection in Controlled Atmosphere (GERPAC) [34]. A variation of concentration outside the 90–110% range of initial concentration (including the limits of a 95% confidence interval of the measurements) was considered as a sign of instability. The presence of BPs and the variation of the physicochemical parameters were also considered a sign of tacrolimus instability. For the sake of comparison, they were also checked against quantities found in a commercial tacrolimus injectable solution (Prograf^®^ 5 mg/mL, Astellas Pharma, Levallois Perret, France). The observed solutions must be of unchanged aspect with regards to the initial aspect. Because there are no standards that define acceptable pH or osmolality variation, pH measures were considered acceptable if they did not vary by more than one pH unit from the initial value [34], and osmolality results were interpreted considering clinical tolerance of the preparation.

### 2.8. Complementary Study: Analysis of a Suspected Leachable Compound

In a complementary study aimed at analysing a suspected leachable compound detected during the simulated use study in the emitted drops, the following experiments were performed. Firstly, to confirm that the detected compound was not linked in any way to the presence of tacrolimus, the silicone part of the Novelia^®^ (blue valve that is in contact last with the eye drop) device was put in contact with 3 mL a KEL-ethanol (tacrolimus free) mixture and left in contact for 10 days at 25 ± 2 °C and 60 ± 5% residual humidity (Binder GmbH, Tuttlingen, Germany) and the solutions were then analysed (*n* = 3). In parallel, in order to know if the extracted compound possessed more of a hydrophilic or hydrophobic affinity, the pieces were put in contact independently with 3 mL of water, 3 mL of an ethanol-water mixture (50/50: *v/v*) and 3 mL of ethanol and left in contact also for 10 days in the same storage conditions and the solutions were analysed by HPLC. Secondly, in order to try to identify the leachable compound, the ethanol solution in contact with the valves was analysed by Gas Chromatography coupled to a Mass Spectrometer (GC-MS) on a Clarus 500 GC–MS chromatograph (Perkin Elmer, Boston, MA, USA) using electronic impact ionization tuned to 70 eV and an Optima 5 Accent, 5% diphenyl 95% dimethylpolysiloxane (30 m × 0.25 μm × 0.25 mm ID) capillary column (Macherey-Nagel, Düren, Germany). The vector gas was helium at a flow rate of 0.8 mL/min. The injection volume was 1 μL and the injector temperature was set at 300 °C. The split ratio was 25:1. A temperature gradient was performed: at t0 min the temperature was fixed at 100 °C and it was increased to 15°/min to 160 °C for 1min, then it was increased to 25 °C/min until 300 °C until the 30 min of analysis. The transfer line was set at 300 °C, the source temperature at 230 °C, and the quadrupole temperature at 150 °C. Electronic impact spectra were recorded in the *m/z* 50 to 500 range for the scan cycles. Tentative identification of observed peaks was performed with NIST library. When necessary, it was confirmed by analysis of the corresponding chromatographic standard solution. The most plausible candidate compounds (2,4-Di-tert-butylphenol, CAS 96-76-4 and 1,3-Di-tert-butylbenzene, CAS number 1014-60-4) were then also tested using the GC-MS and tacrolimus HPLC method and the retention times and spectra were compared with those of the leachable compound.

## 3. Results

### 3.1. Quantification of Tacrolimus: HPLC Method Validation

Tacrolimus presented a retention time of 10.4 ± 0.3 min and a relative retention time of 22.2 (relative to the solvent front), thus being nearly identical to the one found in the method published by Peterka et al. [25], which was of 22.1. The calibration curve used is linear for concentrations ranging from 2.5 to 100 µg/mL, with a mean linear regression equation of Y = 5644.959X − 3552.581, where X is the tacrolimus concentration in µg/mL and Y the surface area of the tacrolimus peak. The intercept was not statistically significantly different from zero, and the mean determination coefficient R^2^ of three calibration curves was of 0.9998. No matrix effect was detected. The relative mean relative bias coefficients were less than 3.0% for the calibration points, except for the 2.5 μg/mL, for which it was 9.0%. The mean repeatability RSD coefficient and mean intermediate precision RSD coefficient were less than 5.0%. The accuracy profile constructed with the data showed that the limits of 95% confidence interval coefficients were all within 5.8% of the expected value, except for the 2.5 μg/mL calibration point, for which the upper range limit was 9.7% (see Appendix A). The limit of detection was evaluated at 1 μg/mL, and the limit of quantification was fixed at 2.5 μg/mL. Figure 1 presents the chromatograms of a blank solution, a solution containing only excipients but no tacrolimus, and a tacrolimus formulation diluted to 50 µg/mL in ACN-water (70:30, *v/v*).

### 3.2. Stability of Tacrolimus in Unopened Multidose Eyedroppers

#### 3.2.1. Physical Stability

Visual inspection and chromaticity measurements

All samples stayed limpid and with a slight yellow tinge throughout the study, for both tested concentrations and conservation temperatures, and there was no appearance of any visible particulate matter, haziness or gas development. For the chromaticity measurements, the initial parameters at day 0 were of 98.79 ± 0.06, −0.30 ± 0.03 and 2.23 ± 0.03 for respectively L*, a* and b* for the 1 mg/mL formulation and 99.71 ± 0.02, −0.11 ± 0.06 and 0.72 ± 0.05 for the 0.2 mg/mL formulation (mean ± standard deviation, *n* = 4). These parameters did not vary significantly throughout the study (see Appendix A).

Turbidity

The initial turbidity was of 10 and 3.5 FNU for, respectively, the 1 mg/mL and 0.2 mg/mL tacrolimus concentration. Throughout the study, turbidity did not vary by more than 0.4 FNU when stored at 5 °C and 1 FNU for the 25 °C and 35 °C storage temperature conditions for the 1 mg/mL concentration. For the 0.2 mg/mL concentration, turbidity did not vary by more than 0.7 FNU, 1.4 FNU and 0.4 FNU for, respectively, the 5 °C 25 °C and 35 °C storage conditions.

Viscosity

The initial viscosity was of 4.97 and 3.56 cP, respectively, for the 1 mg/mL and 0.2 mg/mL tacrolimus concentration. Throughout the study, the viscosity did not vary by more than 0.36 cP for both concentrations and all three storage temperatures.

Micelle size

At day 0, for the 1 mg/mL formulation, the micelles were divided into two populations (size distribution): 99.85% had an average size of 1.96 ± 0.66 nm and 0.15% had an average size of 15.31 ± 6.22 nm, while for 0.2 mg/mL formulation, the micelles formed a single population with an average size of 3.01 ± 1.12 nm. During the nine months of storage, the micelle size did not vary by more than 0.03 nm, 0.03 nm and 0.04 nm for the storage condition at 5 °C, 25 °C and 35 °C, respectively, for the 1 mg/mL eye drops and by more than 0.53 nm, 0.54 nm and 0.61 nm for the storage condition at 5 °C, 25 °C and 35 °C, respectively, for the 0.2 mg/mL eye drops. The micelle size for a tacrolimus-free formulation was similar to that of the tacrolimus formulations.

#### 3.2.2. Chemical Stability

Tacrolimus quantification and BPr

At the beginning of the study, the tacrolimus concentrations were of 1.02 ± 0.02 and 0.20 ± 0.01 mg/mL (mean ± 95% confidence interval) for the 1 mg/mL and 0.2 mg/mL formulations. Throughout the study, the concentrations remained well within the 90–110% concentration range when the formulations were stored at 5 °C (after nine months of storage, tacrolimus concentrations were of 98.80 ± 1.88% and 100.03 ± 0.76%, respectively, for the 0.2 mg/mL and 1 mg/mL formulations), but the concentrations decreased when stored and 25 °C and 35 °C, as presented in Figure 2.

This decrease correlates well with the appearance of multiple breakdown products, especially at 35 °C, but also to a lesser degree at 25 °C (Figure 3). Due to the important decrease in tacrolimus concentrations at 35 °C and the appearance of several degradation products, the analyses were stopped after six months of monitoring for both concentrations stored at 35 °C.

In the initial tacrolimus solution, the impurities specified by the European Pharmacopeia were detected at 220 nm and identified. Table 4 presents the ratio of areas under the curve and the relative retention times of the principal’s impurities and breakdown products found in formulations stored at 5 °C compared to the other storage temperatures. In the formulations stored for nine months at 5 °C, the content of impurity A was found in similar quantities to those found in Prograf^®^ and did not exceed the 0.5%. The amounts of TAC H1 in the formulations were lower than in the Prograf^®^ solution. TAC RI compounds and the unspecified impurity (NS1) were not present in Prograf^®^. However, these compounds were present in the formulations at the initiation of the study. Overall, the appearance of breakdown products and impurities was significantly accelerated by increased storage temperatures.

pH

Initial mean pH was of 5.97 ± 2.05% and 5.56 ± 5.03%, respectively, for the 1 mg/mL and 0.2 mg/mL tacrolimus formulations. As shown in Figure 4, a decrease in pH over time was observed for both concentrations when stored at 25 °C and 35 °C. The drop in pH was more important for the 35 °C and 1 mg/mL conditions. When stored at 5 °C, the pH did not vary by more than 0.04 and 0.08 units, respectively, for the 1 mg/mL and 0.2 mg/mL concentration.

Osmolality

Initial mean osmolality was of 769 and 364 mOsm/kg, respectively, for the 1 mg/mL and 0.2 mg/mL tacrolimus formulation. Regarding the 0.2 mg/mL formulation, osmolality did not vary by more than 4.6% (18.75 mOsm/kg) and 3.16% (11.5 mOsm/kg) from the initial mean osmolality during the nine months of storage at, respectively, 5 °C and 25 °C and no more than 4.26% (15.5 mOsm/kg) during the six months of storage at 35 °C. In a similar way, the osmolality of the 1 mg/mL formulation did not vary by more than 7.06% (54.25 mOsm/kg) and 7.25% (55.75 mOsm/kg) of the initial mean osmolality during the nine months of storage at, respectively, 5 °C and 25 °C and no more than 10.8% (83 mOsm/kg) during the six months of storage at 35 °C.

#### 3.2.3. Sterility Assay

None of the four analysed solutions conserved in unopened eyedroppers at day 0, month 3, month 6 and month 9 showed any signs of microbial growth.

#### 3.2.4. Tacrolimus Degradation Kinetics during Storage

Plotting Ln(C), with C in mol/L, as a function of time (seconds), as presented in Figure 5 for the 1 mg/mL formulation, yielded a value of k corresponding to the slope for each storage temperature.

The plot of Ln(k) as a function of 1/*T*, with *T* in Kelvin, was linear (see Appendix A), thus verifying that the reaction obeyed the Arrhenius law. According to Equation (6), the resulting linear regression equation of the three Ln(k) values versus their 1/*T* values was
$$\mathrm{Lnk}=20.429-(11754\times \frac{1}{T})$$

With Ea = 1413.624 J/mol and A = 745005719.4 s^−1^

If a temperature of 30 °C is selected,
Lnk30=20.429−(11754×0.003299) 
Lnk30=−18.34259
$$k30=1.08123\times 10^{-8}\;s^{-1}$$

According to Equation (7), the time to fall to 90% of initial tacrolimus concentration at 30 °C would be
$$t90=\frac{\mathrm{Ln}\left(0.9\right)}{k30}\;$$
t90=9744544.18 s =3.71 months

This information thus allows the estimation of an impact of potential temperature excursions during storage on tacrolimus concentrations. As another example, tacrolimus concentration would decrease by 5% in only five days when stored at 50 °C for the 1 mg/mL formulation (not taking into account other factors like potential physical instability). An estimation of tacrolimus chemical stability during long-term storage can also be made: it would take approximatively 58 months for tacrolimus concentrations to decrease by 5% when stored at 5 °C.

### 3.3. Tacrolimus Concentrations in Eye Drops during Simulated Use

The simulated test was performed on formulations having been stored for eight months (M8) at 5 °C, and the storage temperature was maintained at 5 °C during the test, except for drop emission. No variation exceeding ±10% of the mean concentration measured at M8 was found for the 1 mg/mL eye drops as seen in the Figure 6A. However, for the 0.2 mg/mL concentration, a greater variability in tacrolimus concentrations was observed (Figure 6B). pH and osmolality remained unchanged.

During the simulated test, an additional peak appeared in the 1 mg/mL tacrolimus drops emitted from the eyedroppers. This peak was not detected in the tacrolimus solutions in unopened eyedroppers. Figure 7 presents the chromatograms of a formulation stored unopened for eight months at 5 °C without having been in contact with the Novelia^®^ delivery nozzle versus with the same formulation after contact with the delivery nozzle. Immediately after the first drop, an increase in the AUC of this compound was observed, before it decreased over time to reach a plateau (see Appendix A). The AUC of the compound was higher in the 0.2 mg/mL formulation than in the 1 mg/mL formulation.

This additional compound presented a very similar retention time as one of the breakdown products observed at other storage temperatures and in the 0.2 mg/mL formulation. To ensure that it was not a tacrolimus degradation product, the different parts of the silicone membrane of the ophthalmic multidose device (Novelia^®^ nozzle) were put in contact with tacrolimus-free formulation containing only the excipients and compared with a control solution without silicone membrane. As seen in Figure 8, the compound appeared as soon as the solutions were put into contact with the silicone parts of the multidose device, and this despite the absence of tacrolimus, which indicated that it was not a tacrolimus breakdown product and more probably a leachable compound.

### 3.4. Eye Drop Volume

The mean volume of a drop was determined to be 25.0 µL and 25.7 µL, respectively, for the 1 mg/mL and 0.2 mg/mL formulation, with the density being of 0.9815 and 0.98805.

### 3.5. Complementary Study: Identification of the Leachable Compound

The peak of the leachable compound was detected after 10 days of contact when the silicone parts were put in contact with water-ethanol, ethanol and excipients. This compound can be seen in the absence of tacrolimus, thus confirming that the leachable compound was not linked in any way to tacrolimus. No peaks were seen when silicone parts were put into contact with only water.

Figure 9 shows the chromatograms of the GC-MS analysis obtained from an ethanol solution having been left in contact with the silicone valves for 10 days. Two compounds are clearly visible before 10 min, with retention times of 4.95, 7.43 and multiple compounds between 11.07 min and 17.59 min. Mass analysis of the 4.95 min compound indicated a similarity with 1,3-Di-tert-butylbenzene (CAS number 1014-60-4) or 1,4-Di-tert-butylbenzene (CAS number 1012-72-2), according to the MS NIST library. The compound at 7.43 min was found possibly to be 3,5-Di-tert-butylphenol (CAS number 1138-52-9) or 2,4-Di-tert-butylphenol (CAS number 96-76-4). The peaks detected between 11.07 and 17.59 min seem to belong to the siloxane family. All these peaks presented common *m/z* values of 73, 147, 221 or 207, 281, 355, with a difference of 74 *m/z* units between the fragments.

Analytical standards of 1,3-Di-tert-butylbenzene and 2,4-Di-tert-butylphenol were analysed using the tacrolimus HPLC method and the retention times and UV spectra of these products were compared with those of the leachable compound observed during the in-use assay. 2,4-Di-tert-butylphenol possessed a quasi-identical relative retention time (1.069, relative to tacrolimus) and UV spectrum. In order to support the hypothesis that it is the same compound as that observed during the GC-MS analysis, 2,4-Di-tert-butylphenol was also reanalysed by GC-MS and was found to correspond (see Appendix A).

## 4. Discussion

In this study, we investigated the physicochemical stability of a novel formulation of tacrolimus, at two concentrations (0.2 and 1 mg/mL) at three different storage conditions for nine months (including one month of simulated patient use). For both concentrations, the results showed that the tacrolimus formulation remained stable throughout the study when stored at 5 °C, but tacrolimus degradation occurred when the formulations were stored at 25 °C and 35 °C.

The analytical method used for the quantification of tacrolimus was reproduced from the stability-indicating method published recently by Peterka et al. [25], who performed an in-depth analysis of tacrolimus and its equilibrium compounds, related substances, byproducts and degradation products. In order to transpose this UHPLC method using the HPLC equipment at our disposal, we used a Kinetex^®^ Core-Shell 2.6 μm EVO C18 100 × 3 mm column. The length and stationary phase composition were the same as the column used by Peterka et al., which was an AQUITY UPLC BEH C18 column [25]. However, the granulometry was slightly higher (2.6 μm versus 1.7 μm) in order to deal with the increased precolumn pressure, but the Core-Shell technology [35] allowed the system to maintain an equivalent separation to the one obtained by Peterka et al. [25], but for a longer method. Indeed, the flow rate had to be increased by 0.25 mL/min, and the gradient lengthened in order to obtain an equivalent retention of tacrolimus, with a relative retention time (relative to the solvent front) of 22.24 in our method compared to 22.08 in the original method. Overall, the method used allowed an accurate and precise quantification of tacrolimus and the identification of nine other compounds, including ascomycin (impurity A described in the European Pharmacopoeia). During the forced degradation tests they conducted, Peterka et al. [25] found that tacrolimus was highly labile under alkaline conditions, and relatively stable under acidic conditions. This is coherent with the data reported by Prajapati et al. [36], who found that tacrolimus was most stable in a narrow pH range of 4 to 6.

This information justified the choice of pH for the buffer used in our formulation, ranging from about 5.5 to 6 for the 0.2 and 1 mg/mL formulations, respectively. The physiological pH of tears is around 7.5 [37]; however, it has been proven that their buffer capacity is quite important (with regards to the small drop volume administered), and that the eye can tolerate products over a range of pH values from about 3.0 to about 8.6, depending on the buffering capacity of the formulation [38]. The formulation process used is similar to one previously published for the compounding of high concentration cyclosporine formulations [39]. The tacrolimus powder was solubilized first in ethanol then added to macrogol 35 glycerol ricinoleate before being incorporated into the aqueous buffer, thus forming micelles. Sodium hyaluronate was also included as an excipient because it can help epithelial healing thanks to its lubricating capacity but also allows the migration and proliferation of epithelial cells [40,41]. Ethanol can be toxic for the ocular surface, especially at concentrations higher than 13.8% (138 mg/mL) like those used for ocular surface surgeries such as photorefractive keratectomy or pterygium excision [42,43]. However, much lower concentrations like 2.5% (25 mg/mL) can be tolerated, as shown in a retrospective analysis of 20 mg/mL cyclosporine eye drops containing 25 mg/mL of ethanol, yet even so, 37% of patients treated complained of side effects, the main one being a burning sensation [44,45]. The tacrolimus formulations presented in this work contained even less ethanol (3.76 mg/mL and 18.79 mg/mL for the 0.2 and 1 mg/mL tacrolimus formulation, respectively), representing only 15% and 75% of the ethanol present in the 20 mg/mL cyclosporine formulation, and thus the ophthalmic tolerance should therefore be improved even further, especially for the 0.2 mg/mL formulation.

For both tested concentrations, all parameters were in favour of a physicochemical stability of nine months when stored at 5 °C. Visual aspect, colour, turbidity, micelle size and viscosity all remained unchanged through the study, as did the pH and osmolality of the solutions. Tacrolimus concentrations remained well within specifications and overall related compounds, and potential breakdown products levels remained consistent with levels found at the start of the study. Because tacrolimus is considered as a drug with a narrow therapeutic range, the choice of accepting a 90–110% range of the initial concentration can be debatable. However, the United States Pharmacopoeia allows this interval for a tacrolimus compounded oral solution [46], and other authors have proposed an even larger acceptability range, accepting an unusual 20% of loss of tacrolimus [20]. In our case, when the formulations were stored at 5 °C, tacrolimus concentrations decreased by less than 5%, thus guaranteeing stable tacrolimus concentrations throughout the proposed shelf life. However, during storage at higher temperatures (25 °C and 35 °C), several signs of chemical instability were detected (for both concentrations): decrease in tacrolimus concentrations, increase in related compounds and breakdown products and an acidification of the media (especially for the 1 mg/mL formulation and at 35 °C). This is coherent with other previously published studies. In 2013, a 0.6 mg/mL tacrolimus eyedrop prepared in castor oil and stored at ambient temperature showed a decrease in tacrolimus concentrations after four months of storage [18]. In 2017, Ezquer-Garin et al. published a stability study on a 0.3 mg/mL tacrolimus ophthalmic solution prepared by diluting I.V. tacrolimus ampules (5 mg/mL) with Liquifilm sterile ophthalmic eye drops, stored at three different temperatures (frozen, refrigerated and 25 °C) during 85 days [21]. They showed that tacrolimus concentrations remained stable throughout the 85 days when frozen or refrigerated, but that the percentage of the initial tacrolimus concentration remaining had decreased to less than 90% after 1 month when stored at 25 °C [21]. Recently, Ghiglioni et al. [20] followed the tacrolimus concentrations in a novel ethanol-free formulation they developed, using polyethoxylated castor oil as a solvent for tacrolimus at 1 mg/mL. Even if their published data is difficult to interpret, they reported a loss of tacrolimus of up to 20% after 60 days unopened refrigerated storage [20]. However, neither of these studies followed the pH or reported any breakdown products, possibly because of an insufficiently specific chromatographic method. Two studies investigated the stability of tacrolimus solutions solubilized in 2-hydroxypropyl-ẞ-cyclodextrins [36,47]. As stated previously, Prajapati et al. showed that the tacrolimus had maximum stability between pH 4 and 6 and that hydrolysis was the main cause of degradation in their aqueous media into which they also added various surfactants [36]. The characteristics and stability of the tacrolimus/hydroxypropyl-β-cyclodextrin eye drops were subsequently further evaluated by an excellent study by García-Otero et al. [47]. In this study, the authors followed the tacrolimus concentrations, osmolality, pH and also performed microbiological tests on a 0.2 mg/mL tacrolimus formulation, stored at 4 °C, 25 °C and 40 °C for three months. They also noticed a temperature-dependent degradation, but even when stored at 4 °C, the two formulations they tested lost 10% of the initial tacrolimus in 120 days. This could be explained by the choice of pH (pH 7) which is outside the tacrolimus chemical stability range. Another interesting galenic formulation, using a thin-film hydration method to encapsulate tacrolimus within a chitosan-based amphiphile derivative containing 1 mg/mL of tacrolimus, was also studied [48]. They evaluated the drug concentration, size distribution, zeta potential, pH, osmolarity and viscosity during one month at 5 °C, 25 °C, and 40 °C. After one month, tacrolimus concentrations had decreased by 6%, 59% and 99.97% at those temperatures, but again the initial pH was outside optimum stability range. Interestingly, like in our study, the pH also decreased in the formulations stored at 25 °C and 40 °C, by 0.6 and 0.8 pH units, respectively. This phenomenon could be because of the leaching of acids (such as propionic acid) from the gamma-sterilized low density polyethylene surface of the eyedropper, as was hypothesized during a stability study in the same eyedroppers of an ophthalmic formulation of polyhexamethylene biguanide [49]. Alternatively, the acidification could be caused by the appearance of an acid breakdown product, like the acid proposed by Prajapati et al., possessing a m/z of 844 [M+Na]^+^ [36]. However, as this compound was not referenced by the method developed by Peterka et al. [25], it is possible that it might not be isolated from other compounds (for example, if hidden beneath the impurity A/ascomycin peak, which increased significantly at 35 °C) and thus not detectable as such. Moreover, neither monographies in the USP nor in the European Pharmacopoeia describe this product [26,50]. Because there are no publically available recommendations concerning the acceptable limits of breakdown products in tacrolimus ophthalmic solutions, more studies are needed to be able to evaluate clearly the tolerable limits. Until then, pharmacists and clinicians will have to use their best judgement, taking into account all aspects in order to provide a safe and effective treatment for their patients. The ideal shelf life depends on the intended use of the medication. Most commercialized medications have shelf lives of at least two years; however, for hospital compounded preparation, which is implemented in the absence of a commercial alternative, a shelf life of several months allows the preparation to be compounded in advance, quality checked, stored, transported, dispensed and used by the patients.

This work also used the Arrhenius equation to be able to estimate the impact of various temperatures on tacrolimus degradation. This equation is commonly used to estimate and predict degradation rates at various temperatures for different stability studies of pharmaceutical products [51,52,53]. In fact, even if this method is not explicitly mentioned in the ICH Q1A(R2) guidelines [33], the Arrhenius equation underpins the general principles of these guidelines [54]. However, many precautions need to be taken into consideration for its application to the developed formulation: (1) the Arrhenius equation can only predict chemical degradations and does not consider possible physical instabilities [54], and (2) the used model postulates that no variation of the estimated activation energy will occur over time, which needs to be confirmed. The information that can be computed from its use must therefore be treated as indicative only until consolidated.

During the simulated test, the overall concentrations remained stable, yet variations in tacrolimus concentrations were observed in the emitted drops, especially with the 0.2 mg/mL formulation. This could be attributed to the sorption of tacrolimus to the surface of the silicone parts in contact with the fluid path inside the Novelia^®^ nozzle, similar to what has already been shown for other lipophilic substances like latanoprost, and to a lesser extent, cyclosporine [55]. These variations must, however, be compared to the variable and small quantity actually absorbed by the eye after instillation [56], and they are quite possibly clinically insignificant. Moreover, the appearance of an additional compound in the chromatographic analysis, which was proven to be independent of the presence of tacrolimus and therefore not a degradation product, suggested the presence of a container–content interaction. The variation of concentration could be caused by tacrolimus sorption to the silicone parts composing the Novelia^®^ nozzle and in contact with the fluid path because some studies have reported the sorption of tacrolimus on different types of materials such as PVC [57,58,59] or silicone [60] during parenteral administration. However, in our case, this phenomenon seems variable but limited in intensity. More notable was the appearance of the additional compound that could be associated to a leaching phenomenon. The first analyses on ethanol extracts with GC-MS narrowed candidate substances to 2,4-Di-tert-butylphenol and 1,3-Di-tert-butylbenzene (or corresponding constitutional isomers) through tentative identification with NIST library. Further investigation with corresponding analytical standards showed 2,4-Di-tert-butylphenol to have identical retention time (with both HPLC and GC) and mass and UV spectra to those of the detected leachable compound. This chemical belongs to the phenol antioxidants class used in polymers as do, for example, butylated hydroxytoluene or molecules commercialized under the Irganox^®^ brand. It is used itself or as a precursor for the production of more complex molecules [61]. It is lipophilic (with a LogP of 5.19), thus poorly soluble in water in opposition to organic solvents [62]. Kolliphor^®^ EL and ethanol in the eye drops formulation may have favoured leaching from silicon pieces to the solution. The leaching of 2,4-Di-tert-butylphenol in the solution may possibly be a concern for patients, considering that this compound is currently listed as “under evaluation” for endocrine disruption under EU legislation [63]. However, the quantities the patient would be in contact with would be very low, as the volume of a drop was measured to be of about 20 µL, which in our case is also close to the optimum volume [56]. However, this illustrates once again the importance of checking container–content interactions between medications and the administration devices used, which is a factor very rarely studied in the published literature. Indeed, amongst the previous studies mentioned, only García-Otero et al. evaluated, if only in a preliminary way, the use of the proposed container (polypropylene eyedropper bottles), by determining the squeezing force necessary to dispense a drop, yet without evaluating actual container–content interactions [47]. Upfront anticipation of the type of container is of paramount importance, even more so in ophthalmology, because it could impact formulation (with or without microbial preservatives) and device choice (compatibility of the medication’s active substance or excipients with the materials of the device).

## 5. Conclusions

Tacrolimus ophthalmic micellar solutions at 0.2 mg/mL and 1 mg/mL are physicochemically stable for up to nine months when stored at 5 °C, but additional studies are still needed to evaluate in-depth the container–content interactions that were detected, especially the impact of a compound leaching out of the used eyedropper bottle.

## Figures and Tables

**Figure 1 pharmaceutics-14-00118-f001:**
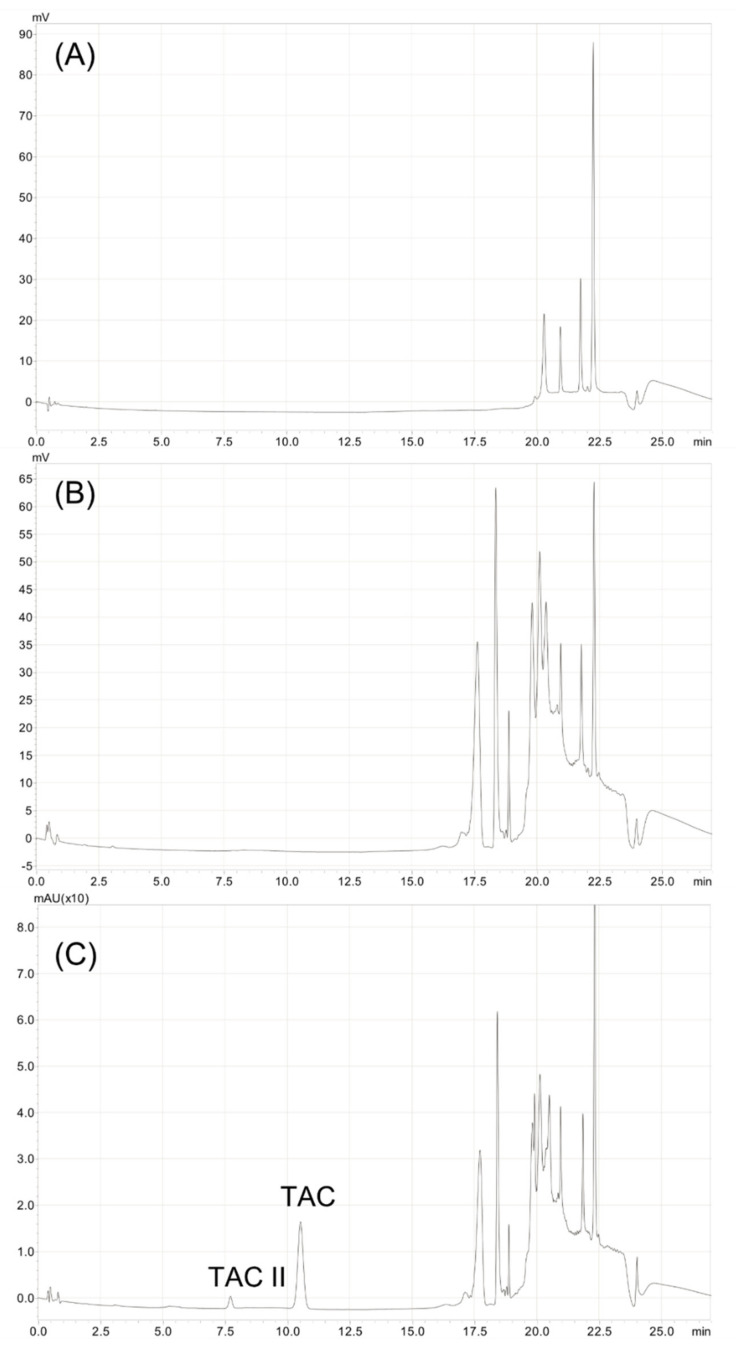
Chromatograms of (**A**) water blank, (**B**) Excipients (formulation without tacrolimus) diluted 1/20th and (**C**) 1 mg/mL tacrolimus formulation diluted 1/20th to 50 μg/mL in acetonitrile-water (70:30, *v/v*). TAC: tacrolimus. TAC II: tacrolimus compound II (equilibrium compound).

**Figure 2 pharmaceutics-14-00118-f002:**
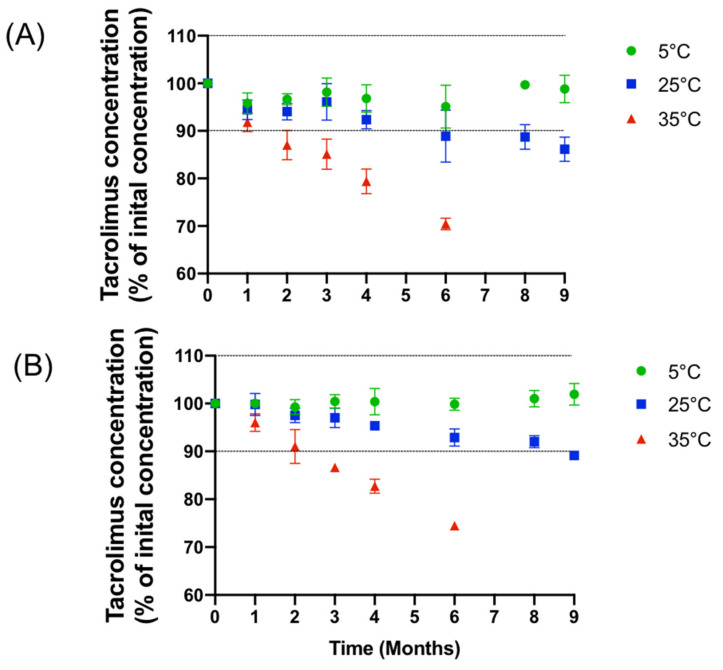
Evolution of the mean concentration of (**A**): 1 mg/mL and (**B**): 0.2 mg/mL formulation with IC 95% during nine months at various storage temperature.

**Figure 3 pharmaceutics-14-00118-f003:**
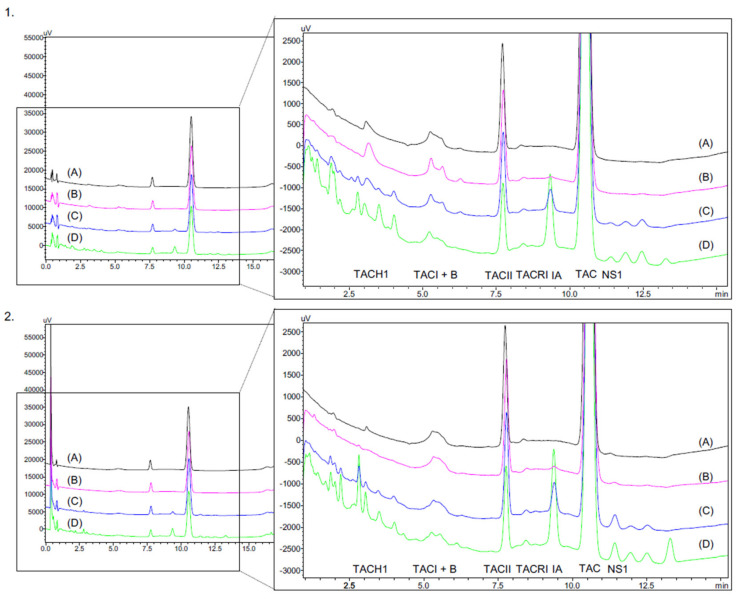
Chromatogram of 1mg/mL (**1**) and 0.2 mg/mL (**2**) tacrolimus formulations at day 0 (D0) (A) and after six months of storage at various temperature: (B) at 5 °C/M6; (C) at 25 °C/M6; (D) at 25 °C/M6. TAC-H1: tacrolimus alpha-hydroxy acid; TAC I: tacrolimus compound I (diol); B: impurity B; TAC II: Tacrolimus compound II (10-epimer) TAC: Tacrolimus; TAC-RI: tacrolimus regioisomer; IA: impurity A (ascomycin); NS: nonspecified compound.

**Figure 4 pharmaceutics-14-00118-f004:**
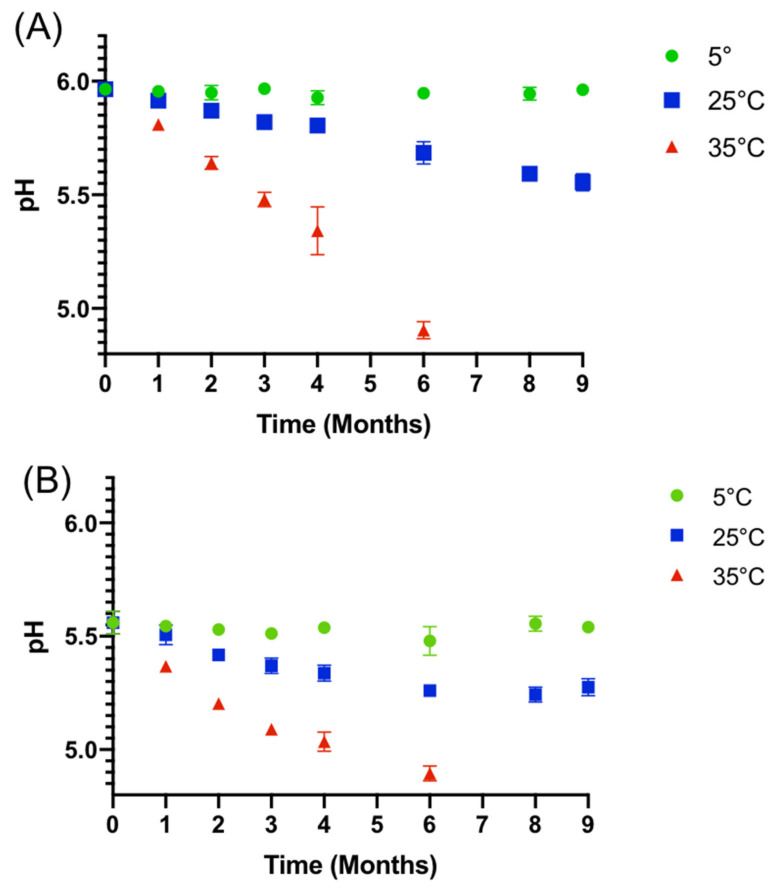
Evolution of the mean pH (±IC 95%) for nine months of storage at various temperatures with (**A**) 1 mg/mL tacrolimus formulation and (**B**) 0.2 mg/mL tacrolimus formulation.

**Figure 5 pharmaceutics-14-00118-f005:**
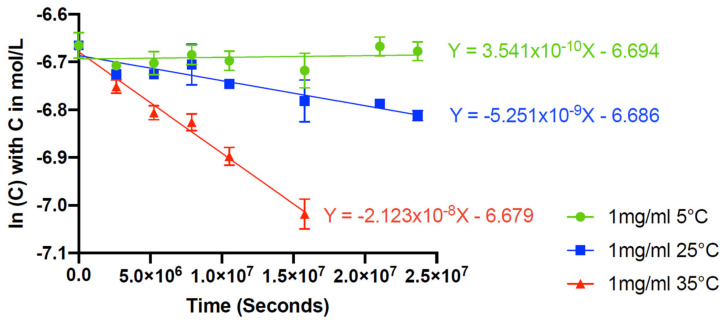
Graphic determination of the rate constant (k).

**Figure 6 pharmaceutics-14-00118-f006:**
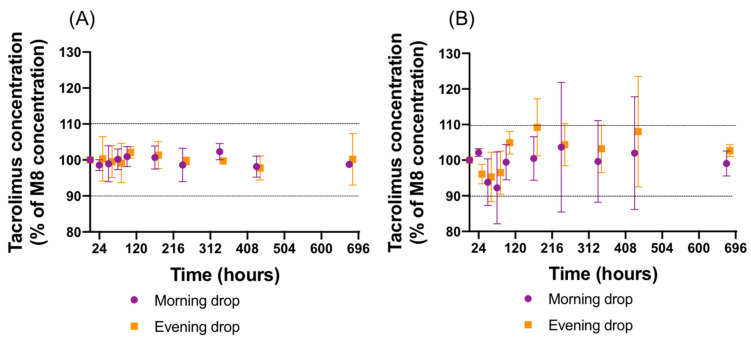
Evolution of the mean concentration (±IC 95%) in opened eyedroppers during one month of storage at 5 °C after eight months of storage at 5 °C with (**A**) 1 mg/mL tacrolimus formulation and (**B**) 0.2 mg/mL tacrolimus formulation.

**Figure 7 pharmaceutics-14-00118-f007:**
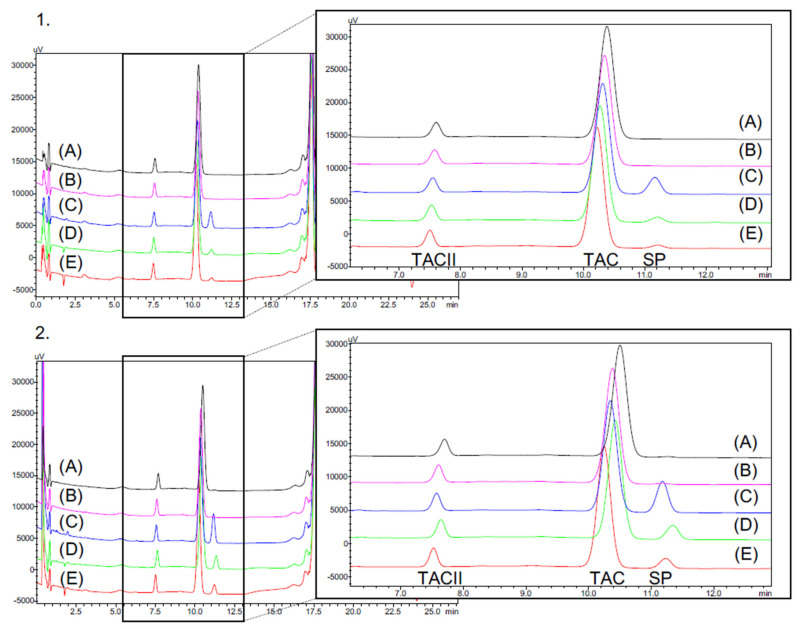
In use assay on the 1 mg/mL (**1**) and 0.2 mg/mL (**2**) formulations with (A): formulation stored eight months at 5 °C, (B): Day 0 = first drop, (C): Day 0 = second drop, (D): Day 14= 28th drop and (E): 56th drop after 28 days of analysis. SP = Supplementary peak.

**Figure 8 pharmaceutics-14-00118-f008:**
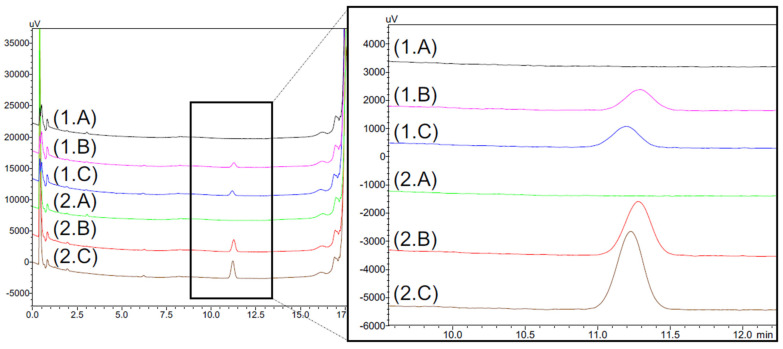
Chromatograms of excipients formulation (TAC-free) of 1 mg/mL (1) and 0.2 mg/mL (2) formulations without contact with the silicone membrane of the ophthalmic multidose delivery system (A) compared with formulations which remained in contact one day (B) and seven days (C) with the silicone membrane.

**Figure 9 pharmaceutics-14-00118-f009:**
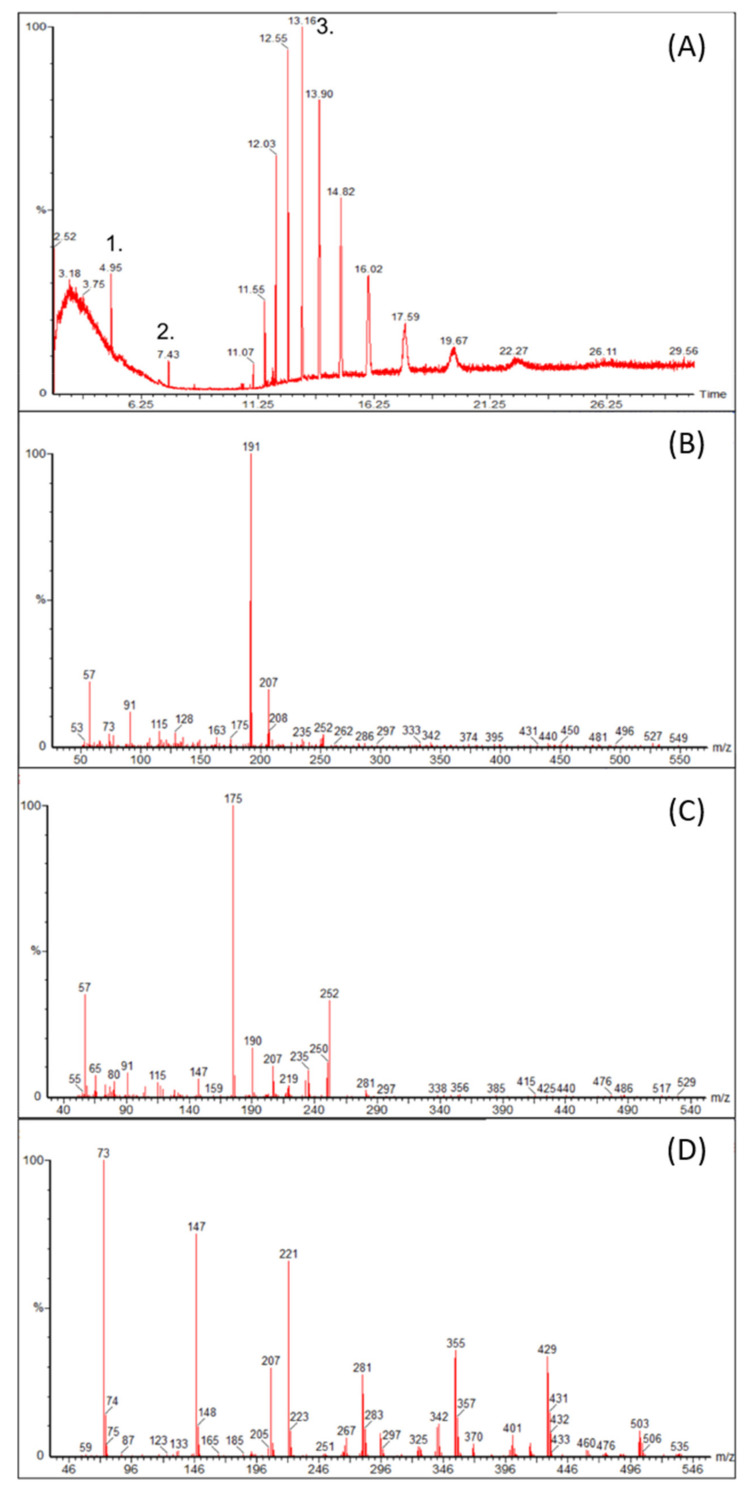
GC-MS analysis of two valves put in contact with absolute ethanol with (**A**) the total scan and (**B**–**D**) the respective *m/z* of the peaks 1. 2. and 3 isolated in the total scan.

**Table 1 pharmaceutics-14-00118-t001:** Composition of the tacrolimus ophthalmic formulations. Q.S: Quantity Sufficient.

Chemical Compounds	Formulation	
	0.2 mg/mL = 0.02%	1 mg/mL = 0.1%
Tacrolimus monohydrate Batch 70312001218, exp 01/10/2023, Inresa, France	200 mg	1000 mg
KEL: Macrogol 35 glycerol ricinoleate (Kolliphor EL^®^) Batch 192835002, exp 30/04/2021, Inresa, France	16 g	80 g
Absolute ethanol Batch 20010089/B, exp 01/24, Cooper, France	4.76 mL	23.81 mL
Buffer solution (composition described below)	Q.S. 1 L	Q.S. 1 L
**Buffer Solution**		
Sodium dihydrogenophosphate dihydrate (NaH_2_PO_4_) Batch 190298040, exp. 30/11/2021, Inresa, France	500 mg	
Disodic monohydrogenophosphate dodecahydrate (Na_2_HPO_4_) Batch 18129611, exp. 30/04/2023, Inresa, France	37 mg	
Hyaluronate sodium Batch PH13560S02, exp 01/12/2023, Inresa, France	1500 mg	
Sodium chloride (NaCl) 0.9% Versylene^®^; Fresenius Kabi France, Louviers, France	Q.S. 1 L	

**Table 2 pharmaceutics-14-00118-t002:** Summary of analyses at each analysis time (X: assessed parameters). TAC: tacrolimus; BPr: breakdown product research.

Studied Parameters						
Months	Visual Aspect, pH, Osmolality, TAC quantification & BPr	Chromaticity	Viscosity	Turbidity	Micelle Size	Sterility Assay
0	X	X	X	X	X	X
1	X			X		
2	X			X		
3	X	X	X	X	X	X
4	X					
6	X	X	X	X	X	X
8	X					
9	X	X	X	X	X	X

**Table 3 pharmaceutics-14-00118-t003:** Gradient used for the liquid chromatography mobile phase.

	Mobile Phase	
Time (minutes)	A (%)	B (%)
0	63	37
1	63	37
12	60	40
17	45	55
19	10	90
22.5	10	90
23	63	37
27	63	37

**Table 4 pharmaceutics-14-00118-t004:** Follow-up of the principal tacrolimus impurities and degradation (expressed as the ratio of the area of the peaks observed over the area of tacrolimus at the initial time and comparison with the impurities found in Prograf^®^. NS: non-specified impurity; RRT: relative retention time, based on tacrolimus retention time, ND: not detected.

Impurities and BP		TAC H1		NS1		TAC RI		Impurity A		NS2	
		%	RRT	%	RRT	%	RRT	%	RRT	%	RRT
Initial condition	1 mg/mL	1.157	0.293	0.110	0.596	0.189	0.794	ND	ND	ND	ND
	0.2 mg/mL	0.180	0.292	ND	ND	0.211	0.793	ND	ND	0.140	1.069
Study endpoint	1 mg/mL 5 °C	0.309	0.299	ND	ND	0.243	0.811	0.230	0.887	ND	ND
	1 mg/mL 25 °C	1.000	0.252	ND	ND	0.259	0.810	3.126	0.887	0.198	1.100
	1 mg/mL 35 °C	1.045	0.265	ND	ND	0.258	0.800	7.263	0.887	0.466	1.083
	0.2 mg/mL 5 °C	0.076	0.269	ND	ND	0.261	0.810	0.328	0.888	0.446	1.098
	0.2 mg/mL 25 °C	1.047	0.269	ND	ND	0.337	0.808	4.251	0.887	1.406	1.099
	0.2 mg/mL 35 °C	2.346	0.267	0.196	0.578	0.374	0.799	9.418	0.887	1.890	1.081
Injectable tacrolimus (Prograf^®^) 5 mg/mL		0.572	0.291	0.050	0.592	ND	ND	0.326	0.892	ND	ND

## Data Availability

Raw data is available in the Appendix A.

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
