# Peer review of "Physicochemical Stability of a Novel Tacrolimus Ophthalmic Formulation for the Treatment of Ophthalmic Inflammatory Diseases"

_pharmaceutics, 2022, doi:10.3390/pharmaceutics14010118_

Round 1
Reviewer 1 Report
The topic of this manuscript is interesting. However, various amendments are still required.
1) The English writing need extensive polishing.
2) The authors declare no conflict of interest. However, is CHU Clermont-Ferrand, Pôle Pharmacie a commercial company? If yes, there is confilct of interest.
3) Is a 9 month shelf-life long enough a an eye drop formulation?
4) What is the degradation mechanism of FK=506?
5) What is the final concentration of ethanol?
Author Response
We thank reviewer for his positive comments that have helped us improve the manuscript. All the suggestions and remarks that were made have been answered or taken into account (changes or items referred to in the answers hereafter appear highlighted in yellow within the revised manuscript).
1) The English writing need extensive polishing.
Answer: Great care was put into the preparation of this manuscript by all the authors, including the corresponding author who is a British national. However, the manuscript was double-checked by another British national, and few minor mistakes were corrected.
2) The authors declare no conflict of interest. However, is CHU Clermont-Ferrand, Pôle Pharmacie a commercial company? If yes, there is confilct of interest.
Answer: “CHU Clermont-Ferrand, Pôle Pharmacie” is the pharmacy department of a public university hospital (or teaching hospital) situated in the town of Clermont-Ferrand, France, and is therefore not a commercial company and does not have any conflict of interest.
3) Is a 9 month shelf-life long enough a an eye drop formulation?
Answer: The ideal shelf life depends on the intended use of the medication. Most commercialized medications have shelf-lives of at least two years, however for hospital compounded preparation, which are implemented in the absence of a commercial alternative, a shelf-life of several months allows the preparation to be compounded in advance, quality checked, stored, transported, dispensed and used by the patients. This information was added to the manuscript (discussion section) lines 679 to 683.
4) What is the degradation mechanism of FK=506?
Answer: Prajapati et al (https://doi.org/10.1016/j.ijpharm.2020.119579) described three main degradation pathways for tacrolimus in an aqueous media: dehydration, hydrolysis (leading to an acid possessing a m/z of 844 [M+Na]+ that could be the cause of the acidification discussed in the manuscript lines 667 to 669) and a combination of dehydration and hydrolysis.
5) What is the final concentration of ethanol?
Answer: the final concentrations of ethanol are of 3.76 mg/mL (0.08 µmol/mL) and 18.79 mg/mL (0.41 µmol/mL) for respectively the 0.2 and 1 mg/mL tacrolimus formulation. This information has been added to the manuscript (Materials and Methods section), line 96 to 98.
Reviewer 2 Report
Generally well written. Lots of data provided. Datas are well discussed.
Minor comments :
Line 135 : "To ensure that a sufficient volume was collected to allow 135 tacrolimus quantification, two drops from two different vials were pooled into one 136container. These two vials remained associated throughout the analysis. Thus, the results 137 of these analyses were returned with n=4."
This line 135 has to be rephrased in order for the reader to fully understand the method applied.
Figure 6 looks awkward, why 1day, 1week, 9days, 13days, etc... ? are these numbers (24,120,312,408,...) randomly chosen ?
Figure 7 legend misses graph 1 and 2 label. I guess Novelia nozzle (1) or not (2).
Author Response
We thank reviewer for his positive comments that have helped us improve the manuscript. All the suggestions and remarks that were made have been answered or taken into account (changes or items referred to in the answers hereafter appear highlighted in yellow within the revised manuscript).
Minor comments :
Line 135 : "To ensure that a sufficient volume was collected to allow 135 tacrolimus quantification, two drops from two different vials were pooled into one 136container. These two vials remained associated throughout the analysis. Thus, the results 137 of these analyses were returned with n=4."
This line 135 has to be rephrased in order for the reader to fully understand the method applied.
Answer: The sentence line 135 has been modified to improved, and now reads “To ensure that a sufficient volume was collected to allow tacrolimus quantification, at each analysis time the drops were pooled by two (each time from the same two vials, which remained associated throughout the analysis). Thus, the results of these analyses were returned with n = 4”. See lines 137 to 140.
Figure 6 looks awkward, why 1day, 1week, 9days, 13days, etc... ? are these numbers (24,120,312,408,...) randomly chosen ?
Answer: the hours indicated in the x-axis of Figure 6 increase by 96 hours (4 days) at every notch after 24 hours.
Figure 7 legend misses graph 1 and 2 label. I guess Novelia nozzle (1) or not (2).
Answer: Indeed, the label of “2.” was missing. The corrected legend of Figure 7 now reads (see line 522):
Figure 7. In use assay on the 1 mg/mL (1) and 0.2 mg/mL (2) formulations with (A): formulation stored eight months at 5°C, (B): Day 0 = first drop, (C): Day 0 = second drop, (D): Day 14= 28th drop and (E): 56th drop after 28 days of analysis. SP= Supplementary peak
Round 2
Reviewer 1 Report
The manuscript has been improved. It appears to be acceptable.
p.s. Is such level of ethanol cause any irritation?
Author Response
Ethanol can indeed be toxic for the ocular surface, especially at concentrations higher than 13.8% (138 mg/mL) like those used for ocular surface surgeries such as photorefractive keratectomy or pterygium excision (see 10.1007/BF02154739 and 10.1167/iovs.13-11717). However, much lower concentrations like 2.5% (25 mg/ml) can be tolerated, as shown in a retrospective analysis of 20 mg/mL cyclosporine eye drops containing 25 mg/mL of ethanol, yet even so 37% of patients treated complained of side effects, the main one being a burning sensation) (see Chast et al, 2004 (10.1016/S0181-5512(04)96181-5) and Kauss Hornecker et al, 2015 (10.1016/j.jfo.2015.02.008). The tacrolimus formulations presented in this work contained even less ethanol (3.76 mg/mL and 18.79 mg/mL, for respectively the 0.2 and 1 mg/mL tacrolimus formulation), representing only 15% and 75% of the ethanol present in the 20 mg/mL cyclosporine formulation, and thus the ophthalmic tolerance should therefore be improved even further, especially for the 0.2 mg/mL formulation.
This information has been added to the discussion section, lines 613-623.